# Grippenet: A New Tool for the Monitoring, Risk-Factor and Vaccination Coverage Analysis of Influenza-Like Illness in Switzerland

**DOI:** 10.3390/vaccines8030343

**Published:** 2020-06-27

**Authors:** Aude Richard, Laura Müller, Ania Wisniak, Amaury Thiabaud, Thibaut Merle, Damien Dietrich, Daniela Paolotti, Emilien Jeannot, Antoine Flahault

**Affiliations:** 1Institute of Global Health, Faculty of Medicine, University of Geneva, 1202 Geneva, Switzerland; laura.muller@unige.ch (L.M.); ania.wisniak@unige.ch (A.W.); amaury.thiabaud@unige.ch (A.T.); thibaut.merle@etu.unige.ch (T.M.); damien.dietrich@icloud.com (D.D.); emilien.jeannot@unige.ch (E.J.); antoine.flahault@unige.ch (A.F.); 2Luxembourg Institute of Health, 1445 Strassen, Luxemburg; 3Institute for Scientific Interchange Foundation, 10126 Torino, Italy; daniela.paolotti@isi.it; 4Addiction Medicine, Department of Psychiatry, Lausanne University Hospital and University of Lausanne, 1004 Lausanne, Switzerland

**Keywords:** influenza, influenza-like illness, syndromic surveillance, participatory surveillance

## Abstract

Implemented in Switzerland in November 2016, Grippenet provides Internet-based participatory surveillance of influenza-like illness (ILI). The aim of this research is to test the feasibility of such a system and its ability to detect risk factors and to assess ILI-related behaviors. Participants filled in a web-based socio-demographic and behavioral questionnaire upon registration, and a weekly symptoms survey during the influenza season. ILI incidence was calculated weekly, and risk factors associated to ILI were analyzed at the end of each season. From November 2016 to May 2019, 1247 participants were included. The crossing of the Sentinel System (Sentinella) epidemic threshold was associated with an increase or decrease of Grippenet ILI incidence, within the same week or earlier. The number of active users varied according to ILI incidence. Factors associated with ILI were: ages 0–4 compared with 5–14 (adjusted odds ratio (AOR) 0.6, 95% confidence interval (CI) 0.19–0.99), 15–29 (AOR 0.29, 95% CI 0.15–0.60), and 65+ (AOR 0.38, 95% CI 0.16–0.93); female sex (male AOR 0.81, 95% CI 0.7–0.95); respiratory allergies (AOR 1.58, 95% CI 1.38–1.96), not being vaccinated (AOR 2.4, 95% CI 1.9–3.04); and self-employment (AOR 1.97, 95% CI 1.33–3.03). Vaccination rates were higher than those of the general population but not high enough to meet the Swiss recommendations. Approximately, 36.2% to 42.5% of users who reported one or more ILIs did not seek medical attention. These results illustrate the potential of Grippenet in complementing Sentinella for ILI monitoring in Switzerland.

## 1. Introduction

Influenza is a serious global health threat that affects an estimated one billion people every year around the world [1]. In temperate regions, the duration and virulence of yearly epidemics depend on the type of influenza virus circulating [2]. Repercussions of these epidemics are dramatic in terms of mortality, hospitalizations, and absenteeism [3]. Every year in Switzerland, influenza is responsible for several thousands of hospitalizations and hundreds of deaths (90% of them among people over 65 years of age) [4]. This disease causes a high burden on the health system. During the 2017/2018 season for example, approximately 209,200 people consulted a primary care physician for influenza-like symptoms (representing 2.5% of the population after extrapolation) [2].

In Switzerland, influenza surveillance is carried out every winter through a national Sentinel network of 150 to 250 physicians (general practitioners, internists or pediatricians) called Sentinella [5]. Similar systems exist in most European countries, collectively named the European Influenza Surveillance Network (EISN) and coordinated by the European Center for Disease Prevention and Control (ECDC) [6]. Each week, Sentinella physicians report clinically-diagnosed influenza cases to the Federal Office of Public Health (FOPH) [7], which allows for the monitoring of influenza-like illnesses (ILIs) in the country. Results are publicly available on the FOPH website every Wednesday of the following week [8]. Each year, the FOPH defines the national epidemic threshold (i.e., the level of ILI incidence above which the influenza season is considered in its epidemic phase), using the moving epidemic method [9], based on data from the last ten seasons [8].

It is important to note that most cases diagnosed by Sentinella physicians do not undergo microbiological confirmation, since this would not alter the treatment course or any other medical action. As such, these cases must be considered suspicions of influenza or ILIs, not influenza cases. An ILI is a subset of symptoms, or a syndrome, that is compatible with a diagnosis of influenza, such as malaise, fever, chills, myalgia, dry cough, loss of appetite, and nausea, as well as a sudden onset of the illness. Several definitions of ILIs exist that use different combinations of symptoms and have different resulting sensitivities and specificities for real influenza cases. Some of these ILIs can be caused by non-influenza viruses, like those responsible for common colds, or even non-viral illnesses.

Sentinella doctors may send nasopharyngeal swabs from patients with a suspected influenza infection to the National Influenza Reference Centre (CNRI) for analysis [10]. For the 2018–2019 season, 40% of the 1001 examined samples were positive for an influenza virus [2]. These analyses also allow the FOPH to determine the circulating viral strains. The Sentinel monitoring of influenza through ILIs has proven its effectiveness despite the absence of systematic microbiological confirmation [11].

Although this surveillance model is very effective, it also has disadvantages. Since not everyone experiencing influenza symptoms seeks medical attention, Sentinella potentially only captures information on a subgroup of the population. Older individuals with risk factors tend to consult more often than healthy young people, who also generally develop fewer complications. Therefore, the number of people affected by the disease might be underestimated by Sentinella, especially among young adults, healthy individuals, as well as those with minor symptoms who do not seek medical help. Furthermore, the time between the beginning of symptoms and the analysis of data by the FOPH can create a significant delay in the detection of the epidemic. The use of a reactive complementary monitoring system to fill these gaps is therefore highly relevant.

Grippenet is a relatively new ILI monitoring system that was implemented in Switzerland during the 2016–2017 influenza season, as a result of a collaboration between the Institute for Scientific Interchange (ISI) Foundation (Turin), the University of Geneva, and the Swiss Federal Institute of Technology of Zurich [12], with the goal to create a reactive monitoring system.

Unlike Sentinella, Grippenet does not rely on physicians but on voluntary citizens to report the presence or absence of influenza-like symptoms through a web portal. Everyone, regardless of age or comorbidity, is invited to participate. Collecting data directly from the population reduces the delay between symptom onset and case recording and permits the inclusion of individuals that do not consult a physician. Grippenet is part of the Influenzanet consortium, which includes 10 countries (Denmark, France, Germany, Italy, Ireland, Portugal, Spain, Sweden, Switzerland, and the United Kingdom). Switzerland joined the consortium in 2016. Member countries all use the same ILI definition, which makes it possible to compare results [13].

Another advantage of this type of participatory surveillance is the inclusion of the general public in scientific research, which fosters interest and health literacy in the population [14]. For example, Grippenet coordinators send weekly email reminders as a way to motivate users to participate, through a newsletter with updates on the most recent preliminary results of the study and the evolution of the Influenza epidemic in Switzerland and in Europe. At the same time, this newsletter is also an opportunity for researchers to disseminate prevention messages and information about influenza.

General information about influenza and its prevention is freely available on the website, as well as a detailed description of the Grippenet monitoring system. The information gathered with the help of participants is also returned to the public through a freely available map on the Grippenet website, showing the incidence of declared ILIs and the number of active users by postcode (see Figure 1). This map is updated every 30 min and summarizes the cumulative incidence of reported ILIs among active users over the last three weeks.

This article aims to evaluate Grippenet's performance in monitoring ILIs in Switzerland, assess what advantages it can provide in comparison with Sentinella, identify risk factors associated with contracting an ILI, evaluate the proportion of vaccinated users, and investigate medical-care seeking behaviors.

## 2. Methods

### 2.1. Participant Recruitment

Participants were recruited through local and national media (TV, radio, printed and online newspapers, blogs) and communication campaigns (social media, university websites, intranets and email lists, advertisements in public transportation in the Geneva area). Individuals can sign up at any time of the year on the following website: http://www.grippenet.ch. The website interface exists in French and German, which are the two most frequently spoken national languages. The only exclusion criterion is that participants must be living within the Swiss territory (a home zip code is required). Participation takes place on a voluntary basis with no financial compensation. Website users can register several individuals on their own profile, allowing them to provide information about young children, elderly people, or individuals who are unable to use a computer.

### 2.2. Data Collection

After registration, participants are invited to fill out an intake online form with questions about age, gender, household size and composition, social contacts, educational level, occupation, primary residence and work zip codes, modes of transportation and commuting time, influenza vaccination, chronic diseases, pregnancy, smoking habits, allergies, diet, and pets (see Appendix A). The answers submitted in this intake questionnaire can be updated by participants at any time. Statistics are computed on the last updated version of the questionnaire, even if it dates back to a previous season. This makes participation easier since there is no need to recreate a new profile each year. There is however one exception: users must specifically update their vaccination status each season for it to be included in the analysis.

During the influenza season (from early November up to May) users receive a weekly newsletter with a link to a second form. This weekly questionnaire assesses the presence of influenza-like symptoms during the previous week, such as fever, cough, sore throat, shortness of breath, myalgia, arthralgia or fatigue. If at least one symptom is declared, further precisions are asked, including date of onset, intake of medication, and medical help seeking behavior (see Appendix A). Symptoms can be reported at any time of the week. The questionnaires can be updated within the same week if necessary, for instance if new symptoms appear.

Both intake and weekly questionnaires were developed by a research team of the London School of Hygiene and Tropical Medicine during the elaboration of the Flusurvey study in the UK, which is part of the Influenzanet consortium [15]. The Grippenet questionnaires are based on the same template used for all Influenzanet countries, with some of the possible answers adapted to Switzerland, for example regarding educational level, medical care options, etc.

Aggregated data from all intake and weekly questionnaires are stored on secure servers (see Section 2.6) and retrieved for every analysis via a remote SSH2 (Secure Shell Client) public key encrypted connection. Once the connection is established, csv files are generated through Python (Python Software Foundation, Beaverton, USA) language functionality and transferred onto a dedicated password-protected computer using the Bitvise (Bitvise Limited, Colleyville, USA) remote access software for Microsoft Windows.

### 2.3. Definitions

#### 2.3.1. Influenza-Like Illness

All Influenzanet participating countries use the ECDC’s ILI definition, which has a sensitivity of 96.1% and a specificity of 6.6% [16,17]. According to this definition, an ILI is defined as the following combination of symptoms:Sudden onset of symptoms;AND (fever or fatigue or muscular/articular pain);AND (coughing or sore throat or shortness of breath).

Several ILI case definitions have been developed for epidemiological surveillance, including definitions from the Centers for Disease Control and Prevention (CDC), the European Centre for Disease Prevention and Control (ECDC), and the World Health Organization (WHO). In addition, some surveillance systems use the acute respiratory illness (ARI) definition, which is less sensitive but more specific [16]. The above ILI definitions generally have a sensitivity between 90% and 96% and a specificity of 7% to 21%. As such, they might miss some influenza cases, but more importantly they include a number of disease episodes not due to influenza, since the specificity is low. Since the goal of influenza surveillance is the identification of epidemic trends and risk factors rather than diagnosis or treatment, these definitions are most often deemed sufficient for these purposes despite the low specificity [11].

The ILI definition used by the Sentinella network is likely more specific because a clinical assessment is conducted during a medical consultation. It is based on the following criteria: sudden fever (>38 °C) and cough or sore throat, possibly accompanied by a marked feeling of sickness or weakness, headache, muscle, joint or generalized pain, and gastrointestinal symptoms. Physicians are also required to report consultations for complications of an influenza infection (pneumonia, bronchitis, ear infections, etc.), if influenza has not already been reported as the initial condition [2].

#### 2.3.2. Active User

In the Grippenet database, a distinction is made between a user and an active user. To be considered a user, a participant must have submitted at least one weekly questionnaire in the past, regardless of the season. To be considered an active user, a participant must have completed at least one weekly questionnaire during the week of the analysis or within the previous 14 days. Furthermore, all participants having reported an episode of ILI are considered active users until the end of the season for statistical purposes.

### 2.4. National Data Sources

To compare the Grippenet cohort to the Swiss population, we gathered data from several sociodemographic and health-related datasets that are described hereafter. The demographic, socioeconomic, and medical data of the Swiss population were provided by the Federal Office of Statistics (OFS) [18]. For the variable “household size”, we only found the absolute number and percentage of households of each size (from 1 person to 6+ persons); thus, 3,762,352 households in total. In order to compare absolute numbers of individuals, we multiplied each percentage by the total count of the Swiss population (8,544,527). With respect to vaccination and chronic diseases, we relied on the most recent available data, which was from 2017 and included people 15 years and older. We did not find any data on pregnant women, pets, and contact with at-risk groups that could be used for the purposes of this study.

Data about diet come from the organization SwissVeg [19]. The Sentinella incidence is available on the Sentinella website [8] and is extrapolated from the number of ILI cases per 1000 consultations at participating physicians’ offices. Both Grippenet and Sentinella incidences are calculated weekly during the whole season. We obtained data on the Sentinella study sample from the FOPH (not publicly available).

### 2.5. Statistical Analyses

ILI episodes are identified based on symptoms reported by users in the weekly questionnaire using the ILI definition by the ECDC (see Section 2.3). The beginning of an identified ILI episode is assigned to the week during which the symptoms began. When symptoms last more than one week, only one ILI is counted, on the condition that the user reports the same symptom onset date each time. In case the user no longer remembers the date of symptom onset, the ILI episode is attributed to the week of the first day of fever or, if they do not remember this either, to the week during which the questionnaire reporting ILI symptoms was sent. Several ILI episodes can be recorded within the same season.

Automated scripts (Python, Python Software Foundation, Beaverton, USA) were used to create variables. For example, a dedicated script was used to derive the weekly incidence from the database and obtain the incidence curves described below. Data was then exported to a Microsoft Excel database. The crude incidence was calculated as the number of ILI cases, as defined by the ECDC, divided by the number of active users, defined as in Section 2.2 above. Descriptive statistics and frequencies were analyzed for all variables. Chi-squared tests were used for descriptive statistics and comparison between variables. A *p*-value of less than 0.05 was considered statistically significant.

A multivariate logistic regression was performed to identify factors associated with ILI status. ILI status was used as the primary outcome. In multivariable models, only those covariates were included which were statistically significant in the univariate analysis.

The map visualizing the ILI incidence was constructed by superimposing a color on each postcode area where active users filled out surveys during the last three weeks. The zone was colored blue if the incidence was 0 and red if it was more than 0, meaning cases of ILI had been declared. If a postcode area did not include any active users, no color was superimposed. A legend was added to each zone, indicating the postcode and the number of active users and ILIs.

### 2.6. Data Safety and Ethical Approval

Grippenet data is handled in accordance with Swiss law on data protection and privacy. Registration to the platform requires a username, an email address, and a password. The list of pseudonymized registered users is kept on the server and thus constitutes the list of users who consented to participate. Data is stored in a highly secure data center located in France. De-identification and confidentiality of the data are guaranteed. Data is exported through an encrypted connection on password-secured, dedicated computers in the offices of the research team. All the extracted data is aggregated.

Data is stored for as long as the project continues. If the project is terminated for any reason, data will continue to be stored for a duration of 5 years before being permanently erased. All participants implicitly consent to participate to the study by registering to the platform and are informed that they can withdraw from the study at any time by sending an email to the Grippenet contact address, in which case their data is deleted permanently via the backend interface. The study protocol was approved by the ethical cantonal board of Geneva (Commission Cantonale d’Ethique et de la Recherche—CCER) with the identification number 2017-01846.

## 3. Results

### 3.1. Participants’ Socio-Demographic and Medical Characteristics

From 7 November 2016 to 28 April 2019, 1247 participants (total users) were included. Demographic, lifestyle, and health-related characteristics of users are described in Table 1.

Participants were mostly located in the French-speaking part of Switzerland (69.5%, 40.7% in the Geneva region). Fewer participants lived in the German-speaking part of Switzerland (28.0%) and even less (2.5%) in Graubünden (Romansch-speaking) or Ticino (Italian-speaking). This can also be visualized in the incidence map, which illustrates the regions where active users are located (Figure 1). By comparison, 22.7% of the Swiss population lives in the French-speaking regions, 63.0% in the German-speaking part, and 8.6% in the Italian- and Romansch-speaking areas [20].

### 3.2. Participation

The evolution of the number of active users allowed us to observe how participation varied throughout the influenza season. The beginning of the season was generally marked by a low number of active users, which increased sharply along with the increase of ILI incidence in Grippenet and Sentinella until the peak was reached. Participation was maintained for as long as the epidemic lasted and decreased along with the downturn of the epidemic (Figure 2).

The crossing of the epidemic threshold defined by Sentinella can serve as a marker of the increase observed at the beginning of each season. This threshold was crossed at weeks 1/2017, 50/2017, and 50/2018 for seasons 2016–2017, 2017–2018, and 2018–2019, respectively. During the 2016–2017 season, we counted a pre-threshold mean of 58 active users vs. 517 users at the epidemic peak in week 2/2017. This number went from a mean of 134 to a mean of 621 during the 2017–2018 season (we calculated a mean from the number of users at the two peaks at weeks 2/2018 and 4/2018) and from a mean of 511 to 615 active users during the 2018–2019 season (peak at week 6/2019).

### 3.3. Epidemic Curves

Even if the data of both systems were computed differently, we observed comparable patterns between Grippenet and Sentinella epidemic curves, especially during the 2018–2019 season (Figure 3). In that season, both curves showed a synchronous pre-threshold phase, post-threshold phase, epidemic peak, epidemic plateau, and final downturn. In each season, the crossing of the Sentinella epidemic threshold at the beginning and end of the season was associated with an increase, and respectively a decrease in the ILI incidence observed by Grippenet during the same week. During the 2016–2017 and 2017–2018 seasons, an increase in Grippenet ILI incidence was observed one week before the epidemic threshold was crossed in Sentinella (at week 49 vs. 50 for the 2016–2017 season and at week 50 vs. 51 for the 2017–2018 season).

For all three seasons, we observed a high incidence in Grippenet during the pre-threshold phase, where the number of active users was still low, thus one or a few declared ILI cases could cause a surge and high fluctuations in the incidence. Interestingly, despite similarities in the epidemic curve patterns, the incidence in Grippenet was 10 to 15 times greater than in Sentinella. Reasons for this are explored in Section 4.3.

### 3.4. Factors Associated with an Increased Risk of ILI

We calculated and adjusted odds ratios for demographic, social, and behavioral factors, several of which were associated with a higher risk of presenting one or more ILIs (Table 2). In terms of demographic factors, the risk of ILI varied according to age and sex. The risk was highest in individuals four years of age or younger. By comparison, lower odds were found in age groups 5–14 (adjusted odds ratio (AOR) 0.6, 95% confidence interval (CI) 0.19–0.99), 15–29 (AOR 0.29, 95% CI 0.15–0.60), and over 65 years old (AOR 0.38, 95% CI 0.16–0.93). Compared with women, men had approximately 20% lower odds of reporting an ILI (AOR 0.81, 95% CI 0.7–0.95). In terms of medical factors, respiratory allergies increased the odds of ILI by almost 60% (AOR 1.58, 95% CI 1.38–1.96) whereas not being vaccinated was shown to multiply the odds of ILI by more than a factor of two (AOR 2.4, 95% CI 1.9–3.04).

The analysis showed no significant association between ILI occurrence and the following social and behavioral factors: contact with more than 10 children or teenagers over the course of the day, contact with more than 10 people aged over 65 over the course of the day, contact with groups of more than 10 individuals at any one time, contact with patients, household size, education, diet, living with pets, main mean of transportation, and daily commuting time. Interestingly, the only significant increase in ILI incidence related to a social factor was found in individuals who reported being self-employed (AOR 1.97, 95% CI 1.33–3.03).

### 3.5. Vaccination Status

As described in the methodology, only participants who had updated their vaccination status during a specific season were considered as vaccinated. No vaccination data were collected during the 2016–2017 season. Vaccination rates in Grippenet users were higher than in the general population (39.9% vs. 13.8%, see Appendix A). However, a large proportion of the Grippenet population for which vaccination was indicated, according to the Swiss recommendations [14], reported not being vaccinated (Table 3). During the 2017–2018 season for example, out of the 110 participants who reported suffering from a chronic disease, only 52.7% (58 individuals) reported being vaccinated. During the 2017–2018 season, the proportion was similar with 53.8% of 65 participants suffering from a chronic disease reported getting a vaccination. The most vaccinated at-risk group were people older than 65 years, during the 2018–2019 season, with a 66% vaccination rate. Among users with at-risk contacts (the variable called “contact risk”: contact with more than 10 children or teenagers over the course of the day, contact with more than 10 people aged over 65 over the course of the day, contact with groups of more than 10 individuals at any one time, contact with patients), only 81 out of 182 (44.5%) underwent vaccination. There were not enough pregnant women in our cohort to yield significant results regarding vaccination in this category. The percentage of users who had an indication for vaccination but had not been vaccinated was similar between the 2017–2018 and 2018–2019 seasons.

### 3.6. Medical Care Seeking Behavior

For the 2016–2017 season, 183 out of 431 users who reported one or more ILIs, said they did not consult a medical doctor (42.5%). This proportion was 37.6% in 2017–2018 (261 out of 695) and 36.2% in 2018–2019 (176 out of 486 participants), for an average of 38.5% individuals not seeking medical care for one or more ILI episodes over all three seasons.

## 4. Discussion

### 4.1. Population

The Grippenet population is not representative of the Swiss population with respect to several characteristics (Appendix A). More individuals are aged between 30 and 64 years in Grippenet than in the general population (62.1% vs. 49.2%, *p* < 10^−4^), whereas individuals 14 years and younger, as well as individuals older than 64 are underrepresented. The Grippenet population tends to be more frequently female (57.3% vs. 50.4%, *p* < 10^−4^) and to have attained a higher level of education (tertiary degree: 62.3% vs. 35.0%, *p* < 10^−4^) than the Swiss population. The Grippenet population also includes larger households, with more participants living in households of 3, 4, 5, 6, or more, and less living alone or with only one person. Grippenet participants seem in better health than the Swiss population, with less than half the number of individuals suffering from a chronic illness (15.6% vs. 32.7%, *p* < 10^−4^). However, the prevalence of respiratory allergies is higher in Grippenet (34.7% vs. 24.0%, *p* < 10^−4^). The Grippenet population also seems to have better health-related habits, with a lower prevalence of smoking (14.5% vs. 27.1%, *p* < 10^−4^) and a vaccination rate more than twice that of the Swiss population (39.9 vs. 13.8%, *p* < 10^−4^).

The age distribution of the Grippenet population is likely affected by patterns of Internet use, where older adults are less Internet-proficient [21]. Likewise, our surveys cannot be taken individually by children in the 0–4 years old group, as well as part of the children in the 5–14 years old group, so these individuals depend on their parents to report them in the system.

As is often the case in volunteer studies, a self-selection bias is likely at play, in that those who took the time to sign up are more health-conscious. Young individuals and children might be less interested in health topics, since they have not yet had much experience with illness, including influenza. In addition, women have been shown to be more interested than men in health topics and seek out health information more actively [22].

Part of our recruitment was done through health-related media broadcasts, which would preferentially reach individuals who value health. This may be one of the reasons why we see a higher proportion of vaccinated users in Grippenet. Another communication channel we used was university information networks. This can contribute to explaining the overrepresentation of female individuals, participants of working age, and with a higher educational level in our sample, as this fits the profile of the average person at most academic institutions.

Differences in health-related characteristics between Grippenet participants and the Swiss population could be partly explained by a reporting bias. It is possible that, knowing the study is led by a public health research institution, participants felt compelled to declare vaccination against influenza and hide smoking habits. However, the anonymity of participation and the fact that questionnaires are self-administered without the need of face-to-face contact with a health professional limits the risk of this bias. Similarly, even though data is never divulged to third parties, individuals might be hesitant to declare health issues, by fear they could be traced back to them.

Allergies in Switzerland have been shown to be more prevalent in young adults, individuals with a higher educational level, and women [23]. This could explain the higher prevalence of respiratory allergies in Grippenet compared to the general population, as these three socio-demographic categories are overrepresented in our cohort.

In terms of geographic distribution, our recruitment efforts were mainly concentrated around the Geneva lakeside and French-speaking Switzerland, due in part to the fact that our research team is based at the University of Geneva. The geographical distribution of our participants mirrors this. Since Switzerland has four languages and many regional and cultural differences, it cannot be excluded that our results would have been different if our data represented the country more homogeneously. Further, influenza epidemics do not reach all areas of the country at the same time. Whereas Sentinella has a nationwide sample at its disposal, ours mainly covers the southwestern corner of the country. This might make the comparison between Grippenet and Sentinella incidences less reliable.

It is interesting to note that all Influenzanet member countries experienced the same representativeness difficulty to a level. Despite this fact, participatory monitoring systems have been shown to parallel the data gathered by Sentinel Systems [24,25,26,27].

Grippenet aims to provide information on populations less represented in Sentinella by using a different methodology. As seen in the comparison of Grippenet and Sentinella populations (Appendix A), Grippenet includes healthier individuals and people at lower risk of complications, while Sentinella has more elderly individuals and children. This can be explained by two hypotheses: First, children are more represented in Sentinella because the network comprises pediatricians, and children usually need to consult more often than young and middle-aged adults. Second, older adults are also more represented in Sentinella since they have a higher probability of seeing their physician in case of an episode of ILI. This is because they tend to have more comorbidities for which they are regularly seen by a general practitioner and are at higher risk of complications (due to comorbidities or to age itself). Conversely, as shown above, the ages in between and people without health issues are the ones that are most represented in Grippenet. This difference in the population makes the two systems complementary for ILI monitoring and research.

### 4.2. Participation

As shown in the results, the Grippenet participation rate varies over the course of a season, mirroring the evolutionary phases of the influenza epidemic. The rapid increase in active users at the beginning of each influenza season might reflect more intensive efforts on our part to recruit participants during this period, while after the start of the epidemic, the main goal shifts to maintaining a regular participation until the end of the season. The increase in active users might also be the sign of a renewed interest in Grippenet from the public, as attention is drawn to influenza during that period of the year by the media and people around us suffer more frequently from upper-airway respiratory illnesses.

The accuracy of outbreak monitoring in a system like Grippenet is highly dependent on the number of participants. Studies have shown that with an active participation rate of 0.5 individuals per 10,000 inhabitants, the Influenzanet surveillance model is already capable of monitoring an influenza epidemic [15,24,27,28,29,30]. With a population of 8.57 million, this represents a little under 500 active users in Switzerland. Grippenet was able to reach this number in the 2017–2018 and 2018–2019 seasons, except for the pre-threshold phase (Figure 2). During the 2019–2020 season, 538 users were active within the first week. This number progressively increased over the season to reach 691 active users on week 8/2020. Later on, we were able to use the beginning of the COVID-19 epidemic in Switzerland as an incentive for recruiting even more participants and reached a total of 2500 users and 1412 active users as of week 15/2020.

### 4.3. Epidemic Curves

As shown by this study and other Influenzanet member countries, data collected by a web-based participative surveillance system can be used to track an influenza outbreak [24,26,27,31,32]. For the 2018–2019 season, except in the pre-threshold phase, the overall pattern in epidemic curves in Sentinella and Grippenet was similar.

Although the epidemic curve patterns are similar, the incidence of ILIs is 10 to 15 times greater in Grippenet than in Sentinella. Several Influenzanet member countries described a similar phenomenon on various levels of magnitude [25,31,32,33]. A hypothesis could be that this difference is due to health care-seeking behaviors. Indeed, an individual who chooses not to consult a physician for an ILI will not be counted as a case by Sentinella [32].

The higher Grippenet incidence could also be explained by the very low specificity of the ECDC’s ILI definition, which has a high sensitivity of 96.1% but a specificity of only 6.6% [16]. Indeed, even if Sentinella also monitors ILIs, Grippenet’s capacity to discriminate between an influenza virus illness and a common cold is probably lower in comparison, due to the lack of examination by a medical professional. We do not have sensitivity and specificity estimates for Sentinella’s ILI definition, but during the 2018–2019 season, 40% of the 1001 nasopharyngeal swabs from ILI patients detected by Sentinella physicians tested positive for influenza [2].

The propension of Grippenet to include common colds in the ILI incidence is further evidenced by the fact that, during the 2018–2019 season, 488 ILI episodes were reported by 215 users, which represent a mean of 2.3 ILIs per person declaring symptoms. Yet, the probability of being infected with influenza twice in the same season is low. On the other hand, it is frequent to experience several common colds during one season. Finally, studies have suggested that participants of voluntary-based surveillance platforms might feel more motivated to complete surveys on weeks when they have symptoms [15], which may lead to an over-evaluation of the incidence.

The early detection of the onset of an influenza epidemic is another advantage of the Influenzanet system [25,32,33,34]. During the 2017–2018 and 2018–2019 seasons, Grippenet anticipated the detection of the influenza epidemic by about one week. However, as mentioned in the results, Grippenet’s epidemic curve has artificial fluctuations and a high incidence in the pre-threshold phase, most likely due to a low number of active users, which causes an artificial surge in the incidence whenever a few ILI episodes are reported. As discussed above, the results of Grippenet are not reliable until the minimum number of 500 active users is reached.

In contrast to Sentinella, which has established an epidemic threshold based on the last 10 seasons, Grippenet has yet to determine one. In the future, establishing an epidemic threshold for Grippenet would facilitate the early detection of an influenza epidemic. The moving epidemic method (MEM) could be used for this purpose. The MEM establishes an epidemic threshold based on historical data: the more historical data are available, the more accurate this method [9]. Several studies showed that the MEM could be a robust and flexible method to allow for an evidence-based assessment of seasonal influenza activity, sometimes after as few as four seasons [35,36,37]. Importantly, participation in Grippenet needs to be consistent before and during the beginning of the season to identify the threshold with accuracy.

### 4.4. Risk Factors Associated with the Occurrence of an ILI

In terms of demographic factors, young children, people of working age, and women were at higher risk of ILIs, as were individuals suffering from respiratory allergies. In the literature, there is no solid evidence about factors associated with a higher risk of presenting an ILI. Age older than 65 years and chronic conditions are typically correlated with a higher rate of complications, rather than a higher incidence [2]. In our cohort, during the 2018–2019 season, vaccination was associated with a 2.4-fold decrease in the risk of ILIs. Effectiveness of the influenza vaccination varies according to the viral strains in circulation and composition of the vaccine, as well as other individual factors, but is usually described as 70% to 90% effective [38].

Regarding socio-behavioral factors, the only significant correlation we found with ILI incidence was the reporting of self-employed status. We did not find any results in the literature consistent with this finding. One hypothesis could be that stress levels are high in this professional group and have a negative impact on immunity. However, considering the small number of individuals in this category, this might be a fortuitous result and analysis should be repeated on larger samples before confirming. It is highly likely that other risk associations related to socio-behavioral aspects exist; however, our sample size from these past three seasons was too small to yield significant results for such subcategories.

### 4.5. Vaccination Status

The WHO recommends that influenza vaccination coverage be at least 75% in at-risk groups and healthcare professionals [39]. Even though we showed that our participants have a higher vaccination rate than the general population, this rate is still insufficient in at-risk groups, with 51.2% of all at-risk users vaccinated in 2018–2019.

Grippenet participants with a positive response to the “contact risk” variable are likely to be healthcare professionals or to take care of children or elderly individuals. The FOPH estimates that around 23% healthcare professionals (in contact or not with patients) are vaccinated [2]. In our study, this number is higher but still insufficient (44.5% in 2018–2019).

### 4.6. Medical Care-Seeking Behavior

Sentinella estimated that 2.5% of the Swiss population consulted a primary care physician for an ILI during the 2018–2019 season [2]. This figure does not reflect the actual proportion of the population who have suffered from influenza. Our study showed that between 36.2% and 42.5% of the Grippenet participants who reported at least one ILI did not seek medical attention. The Grippenet surveillance model has the advantage of including people who do not seek medical help, in contrast to Sentinella, which functions through primary care physicians. This can constitute a complementary tool to evaluate the burden of influenza.

### 4.7. Strengths and Limitations

This study is the first in Switzerland to evaluate a participative approach to the surveillance of seasonal influenza epidemics. It is also one of the few studies to look at demographic, medical, and socio-behavioral factors associated with the risk of ILIs. Participation in Grippenet is easy, quick, and free. By bypassing physicians, Grippenet may reduce the time necessary to capture the changes in ILI incidence and has the potential for real time analyses. Data provided by users is currently analyzed weekly, but we hope to implement daily analysis soon, as is already done in France, for example [40].

In a crisis, Grippenet can easily and rapidly be adapted for use in the context of an emerging infectious pathogen, as was the case during the COVID-19 epidemic. Indeed, the weekly analysis allowed us to detect an unexpected rise in incidence during the 10th week of 2020, near the habitual end of the influenza season, corresponding to the beginning of the novel coronavirus epidemic in Switzerland. From that point onward, the Grippenet platform has been adapted, particularly in terms of new items added to the weekly questionnaire, to better follow the COVID-19 epidemic. Additionally, questions regarding social and behavioral patterns were added to the weekly questionnaire, allowing for the evaluation of the population’s response to the preventive measures put in place by public health authorities.

Finally, the newsletter sent every week can be used to sensitize the population to the importance of preventive measures such as hand hygiene and vaccination, and emphasize key messages carried out by Sentinella physicians.

This study has several limitations. We already mentioned the lack of representativeness of the Grippenet population compared to the Swiss population, as well as a sample size too small to analyze the association between the risk of ILIs and certain socio-demographic or medical factors, and to allow for accurate incidence calculations in the first few weeks of the influenza epidemics. However, this last issue is progressively being remedied, as the number of participants increases each year thanks to our communication efforts. As the sample size grows, we could, for example, select a representative subsample and proceed to subgroup analyses.

A potential source of error is the possibility for participants to continue reporting symptoms without updating their demographic questionnaire (with the exception of vaccination status which is not taken into consideration if not updated within the current season). Newly diagnosed diseases (such as asthma or diabetes) and changes in socio-demographic characteristics (change of address, number of children, divorce, marriage) could occur over time and, if the intake survey is not updated, not be taken into account in the analysis. Making the update of the demographic questionnaire mandatory once or twice a year could help remedy this, but on the other hand might deter users from continuing their participation, since the intake survey is relatively long compared to the weekly questionnaire.

Our results are highly dependent on definitions chosen. We decided to define active users as users who completed at least one weekly questionnaire during the previous 14 days or the week of the analysis or presented one or more ILIs during the current season. Other Influenzanet member countries chose to define active users as users who filled out the survey inside a participation window of ±1 week around the week of the analysis. These two definitions result in different numbers of active users and therefore in a different ILI incidence. Thus, it would be helpful for Influenzanet countries to agree on a harmonized definition, in order to reliably compare results between countries.

## 5. Conclusions

This study shows that a participatory influenza monitoring system such as Grippenet can help monitor ILI cases in a fast and flexible way, identify ILI risk factors and gaps in the influenza vaccination coverage, as well as analyze medical care-seeking behaviors.

A key element of Grippenet's success is the active participation of a large cohort of voluntary citizens. Some of the advantages of this platform are its simplicity and rapidity. Registering is easy, there are no exclusion criteria for individuals living in Switzerland, and a few clicks suffice to participate. This also makes it faster to gather and analyze data than with traditional surveillance systems. Another advantage of Grippenet is its flexibility to adapt to other pathogens, especially emergent viruses, as has been done recently with the real-time monitoring of the COVID-19 epidemic in Switzerland. In the future, Grippenet could even be used to evaluate vaccination efficacy, as has been done in France [41].

Grippenet's main challenge for the future is to recruit more participants outside the French-speaking region of Switzerland and in certain categories (individuals younger than 15 and older than 65, with a lower educational achievement, suffering from chronic diseases, smokers) in order to achieve a larger and more representative sample and thus allow for subgroup analyses (for example, by geographical location, age or gender).

Grippenet does not claim to replace the Sentinel surveillance model but has the potential to enhance it by collecting information in real time, including from a different population profile and individuals that do not seek medical help, as well as promoting health in Switzerland through information spreading, awareness raising, and partnership with citizens and communities.

## Figures and Tables

**Figure 1 vaccines-08-00343-f001:**
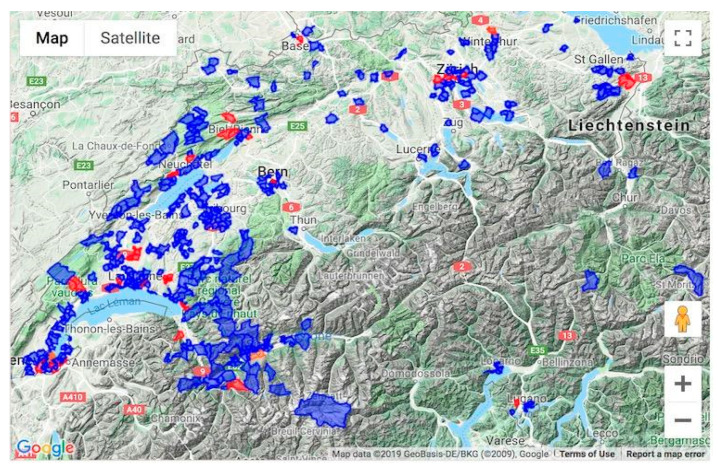
Geographical distribution of active users at the end of 2018–2019 season. Blue: areas with at least one active user; Red: areas with at least one influenza-like illness (ILI) case.

**Figure 2 vaccines-08-00343-f002:**
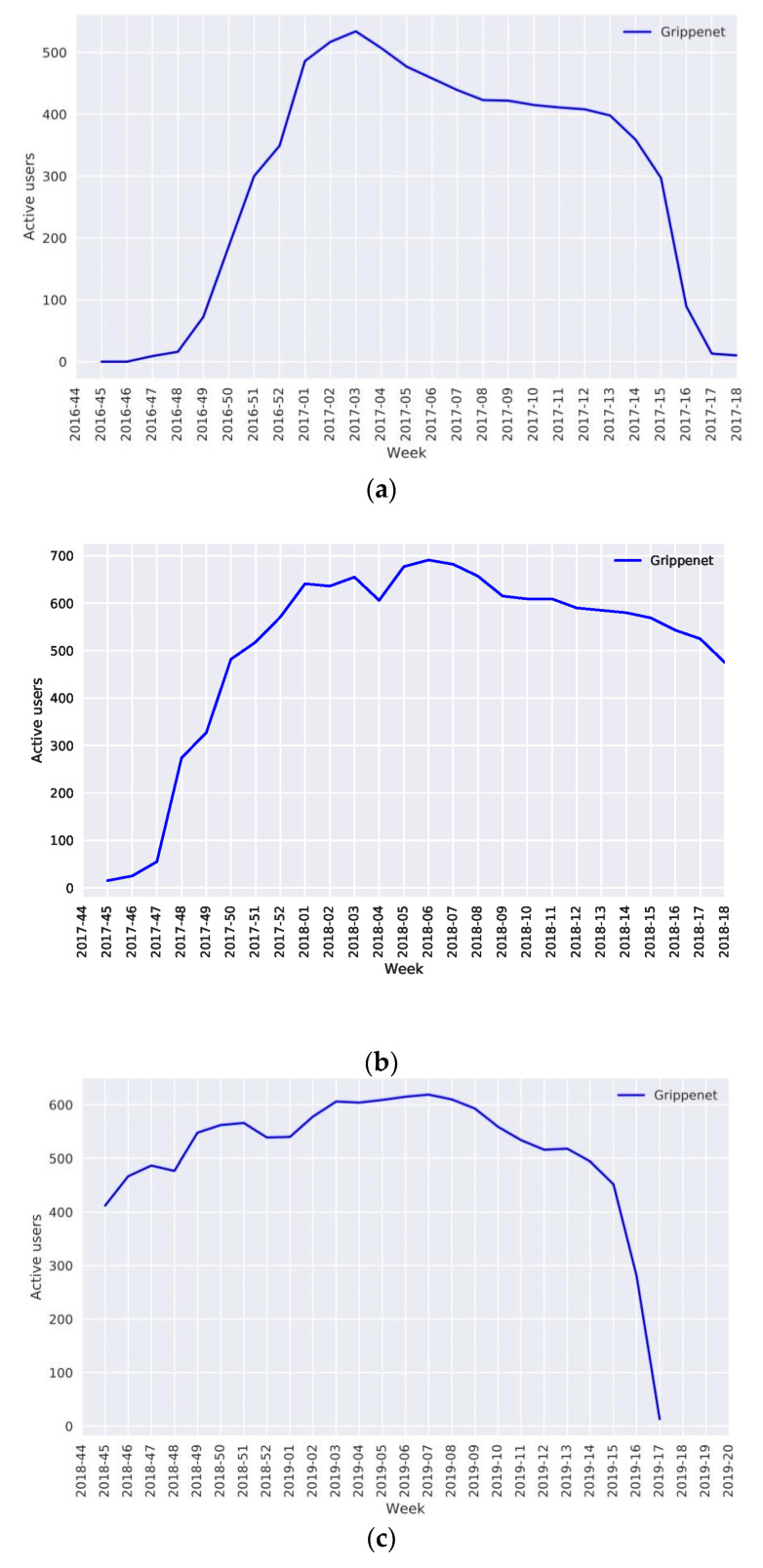
Evolution of the number of active users during seasons: (**a**) 2016–2017, (**b**) 2017–2018, (**c**) 2018–2019.

**Figure 3 vaccines-08-00343-f003:**
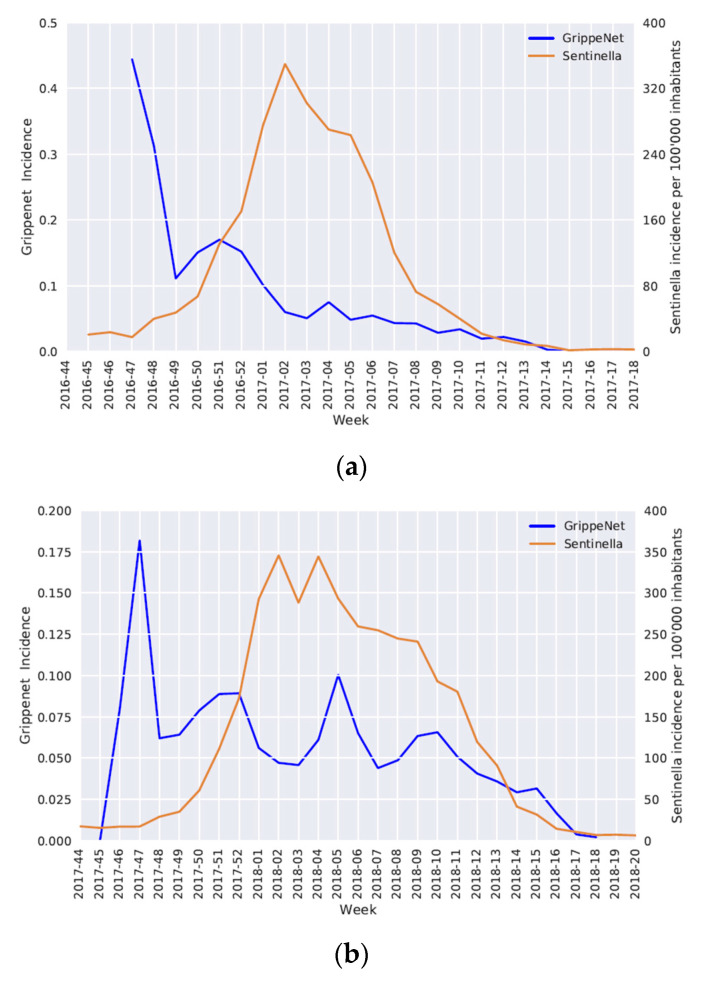
Comparison of Sentinella and Grippenet influenza-like illness epidemic curves during seasons: (**a**) 2016–2017, (**b**) 2017–2018, (**c**) 2018–2019.

**Table 1 vaccines-08-00343-t001:** Medico-demographic profile of the Grippenet population.

Variable	Category	*N*_tot_ = 1247	
		*N*	%
Age (years)	0–4	30	2.4
	5–14	68	5.5
	15–29	207	16.6
	30–64	770	61.7
	65+	164	13.2
	No answer	8	0.6
Sex	Female	714	57.3
	Male	533	42.7
	No answer	0	0.0
Smoking	Yes	165	13.2
	No	971	77.9
	No answer	111	8.9
Respiratory allergies	Yes	395	31.6
	No	744	59.7
	No answer	108	8.7
Chronic medical condition(s)	Yes	178	14.3
	No	961	77.1
	No answer	108	8.7
Influenza vaccination 2018/19	Yes	364	29.2
	No	549	44.0
	No answer	334	26.8
Pregnancy (women only)	Yes	3	0.4
	No	365	51.1
	No answer	346	48.5
Contact risk *	Yes	728	58.3
	No	411	33.0
	No answer	108	8.7
Completed Education	No former education	60	4.8
	Secondary school	354	28.4
	Tertiary degree	635	50.9
	No answer	198	15.9
Main activity	Paid employment, full time	419	33.6
	Paid employment, part time	275	22.1
	Self-employed	77	6.2
	Attending daycare/school/college/university	201	16.1
	Home-maker, unemployed, long-term sick-leave or parental leave	75	6.0
	Retired	162	13.0
	No answer	38	3.0
Household size **	1	241	19.3
	2	294	23.6
	3	192	15.4
	4	246	19.7
	5	86	6.9
	6+	45	3.6
	No answer	143	11.5
Diet	No special diet	986	79.1
	Vegetarian	59	4.7
	Vegan	10	0.8
	Low-calorie	26	2.1
	Other	54	4.3
	No answer	112	9.0
Presence of a pet	Yes	149	11.9
	No	986	79.1
	No answer	112	9.0

* Contact risk: contact with more than 10 children or teenagers over the course of the day, contact with more than 10 people aged over 65 over the course of day, contact with patients or contact with groups of people (more than 10 individuals at any one time). ** Household size: number of individuals living in a household.

**Table 2 vaccines-08-00343-t002:** Risk factors attributed to an ILI episode among all Grippenet users (2016–2017 to 2018–2019 seasons). CI = confidence interval; AOR = adjusted odds ratio. Significant results in bold.

		ILI (*N*)	Non-ILI (*N*)	OR	95% CI	AOR	95% CI
Age group (years)	0–4	20	10	1	-	1	-
	5 to 14	30	38	**0.4**	**0.17–0.97**	**0.6**	**0.19–0.99**
	15–29	70	137	**0.26**	**0.12–0.57**	**0.29**	**0.15–0.60**
	30–64	375	395	0.48	0.22–1.03	0.49	0.23–1.05
	65+	75	89	**0.42**	**0.19–0.96**	**0.38**	**0.16–0.93**
Gender	F	379	335	1	-	1	-
	M	195	338	**0.5**	**0.4–0.64**	**0.81**	**0.7–0.95**
Respiratory allergy	Yes	219	176	**1.38**	**1.08–1.76**	**1.58**	**1.38–1.96**
	No	353	391	1	-	1	-
Chronic medical condition(s)	Yes	99	79	1.29	0.94–1.8	-	-
	No	473	488	1	-	-	-
Smoking	Yes	84	81	1.04	0.79–1.44	-	-
	No	486	485	1	-	-	-
Influenza vaccination 2018/2019	Yes	140	225	1	-	1	-
	No	310	239	**2.09**	**1.59–2.73**	**2.4**	**1.9–3.04**
Contact risk	Yes	378	350	1.21	0.95–1.54	-	-
	No	194	217	1	-	-	-
Household size	1	126	115	1	-	1	-
	2	147	147	0.92	0.65–1.28	-	-
	3	109	83	1.04	0.74–1.44	-	-
	4 or more	176	201	1.12	0.82–1.75	-	-
Education	No former education	28	32	1	-	1	-
	Secondary school	186	168	1.27	0.73–2.19	-	-
	Tertiary degree	311	324	1.09	0.65–1.86	-	-
Diet	No special diet	492	494	1	-	1	-
	Vegan, vegetarian	33	36	0.92	0.57–1.55	-	-
	Low-calorie, other	47	33	1.43	0.9–2.28	-	-
Pet(s) at home	Yes	80	69	1	-	1	-
	No	492	494	0.85	0.60–1.21	-	-
Main occupation	Paid employment, full time	186	233	1	-	1	-
	Paid employment, part time	141	134	1.32	0.98–1.79	1.21	0.87–1.68
	Self-employed	44	33	**1.67**	**1.03–2.73**	**1.97**	**1.33–3.03**
	Attending daycare/school/college/university	86	115	0.94	0.67–1.31	0.98	0.72–1.35
	Retired, homemaker, unemployed, long-term sick leave or parental leave	105	132	0.99	0.72–1.37	1.06	0.79–1.42
Main mean of transportation	Walking	78	74	1	-	1	-
	Bike	61	64	0.9	0.56–1.45	-	-
	Car	247	220	1.06	0.74–1.54	-	-
	Public transportation (bus, train, metro, etc.)	171	190	0.85	0.58–1.25	-	-
Daily commuting time	No time at all	324	289	1	-	1	-
	0–30 min	117	139	0.75	0.56–1.01	-	-
	30 min–1.5 h	99	114	0.77	0.57–1.06	-	-
	1.5 h and over	32	25	1.18	0.68–2.03	-	-

Significant results in bold.

**Table 3 vaccines-08-00343-t003:** Vaccination rate according to indication during 2017–2018 and 2018–2019 seasons.

Indication to Vaccination	Vaccinated Users	
	2017–2018	2018–2019
Age > 65	54.2% (77/142)	66.2% (47/71)
Contact Risk	37.3% (135/362)	44.5% (81/182)
Chronic Disease	52.7% (58/110)	53.8% (35/65)
Pregnancy	0% (0/4)	50% (1/2)
All users (with updated intake)	37.4% (279/746)	47.8% (165/345)
All at risk users	43.7% (270/618)	51.3% (164/320)

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
