# Peer review of "Grippenet: A New Tool for the Monitoring, Risk-Factor and Vaccination Coverage Analysis of Influenza-Like Illness in Switzerland"

_vaccines, 2020, doi:10.3390/vaccines8030343_

Round 1
Reviewer 1 Report
Summary: The authors analyzed and explained the performance of Grippenet, which is a framework that can be effectively monitored for influenza-like-illnesses (ILI) disease, compared with Sentinel, which is another existing framework, to prove its performance.
Strong points: The authors suggested efficient monitoring frameworks for influenza viruses. In particular, The authors have carried out a new fashion research topics in terms of COVID-10 related issues.
Weak points: 1) the authors need to explain a detailed description of data collection and processing step. 2) the table creation is required for data-wide variables. 3) also need to add a description of the painting
(e.g. a-2017-18/ b-2018/19 ...). 4) need to explain and justify the techniques used in the analysis
(basic concepts + why techniques were used). 5) in the paper, the expreimental results should be interpreted in medical and biological meaning. 6) The authors need to explain how determine the threshold in the Sentinel framework and the grippenet framework.
Syntax and typo: there are lots of errors, for examples, (p42 : systems exists >> systems exist, p95 : help seeking >> help-seeking, p107 : for instance >> for instance, p113 : advertisement >> advertisements, p114 : are >> is, p152 : was >> were), and so on. there are too much mistake.
Comments: As a monitoring framework for infectious diseases that has recently become an issue, this paper proposes a new tool called Gripenet. Comparing the advantages and disadvantages of various aspects with the existing Sentinel used. However, the comparative analysis of the monitoring framework presented in this paper only compares the results, and users are not sure what principles these two frameworks move on. For example, other existing infectious disease surveillance frameworks are written how monitoring is conducted by some statistical methods or ml-based method. But, this paper simply compares the differences of existing frameworks as results. (A Simulation-Based Study on the Comparison of Statistical and Time Series Forecasting Methods for Early Detection of Infectious Disease Outbreaks)
Also, in the description of the data, how the data was organized is generally detailed, but only fragmentary information reader can get in the actual text. Of course, even if the author has attached the text to the appendix, some of them should be shown to make the paper more credible. This is the same with the problem of feature selection.
As described above, it is necessary to describe how the threshold is set up and how the comparison is different from the proposed method. Otherwise, this paper is simply a framework on property extraction, data processing.
The title also describes influenza-related analysis tools, with data focused on ILI, not influenza. If you want to use this paper as its title suggests, even if the results are lower than the comparison tool, it would be good to add influenza-related results and add descriptions of those results.
In addition, comparisons were simply compared in a way that compares the actual situation with previous data, and it seems a little more desirable to compare the performance between the two tools by marking them as a more academic indicator (error rate or accuracy).
Finally, the abstract as summary of the paper as well as most parts in the manuscript should be rewritten.
Author Response
Dear Reviewer,
Thank you very much for your valuable comments on our article originally entitled “Grippenet: a new tool for the monitoring, risk-factor and vaccination coverage analysis of Influenza in Switzerland” (vaccines-805609) submitted as an original article to Vaccines.
We value this opportunity to improve our manuscript and to answer your comments. We used the track change function in Microsoft Word to make the revisions clearly visible.
We thank you for your consideration and look forward for our collaboration.
Reviewer 1’s comments
Comment 1: The authors need to explain a detailed description of data collection and processing step.
Answer 1: Thank you for this useful observation. We reorganized the methods section by moving the participants’ recruitment subsection at the top for more clarity. Then we added information about the questionnaire questions (Track changes version lines 155-158 and 165-167) and development (lines 176-180) and the data retrieving process (lines 181-185)
Comment 2: The table creation is required for data-wide variables.
Answer 2: Thank you for this comment. Unfortunately, we do not understand your meaning.
Comment 3: Also need to add a description of the painting (e.g. a-2017-18/ b-2018/19).
Answer 3: For the sake of clarity, as you asked, we described each graph in figures 2 and 3 by adding letters according to the corresponding season, i.e. figure 2.a. season 2016-2017, etc.
Comment 4: Need to explain and justify the techniques used in the analysis (basic concepts + why techniques were used).
Answer 4: Following this useful remark, we expanded the “definitions” subsection of the methods to explain basic concepts about ILI definitions, their sensitivity and specificity and why we used the concept of ILI instead of influenza illnesses for monitoring (lines 189-198). We also expanded the analysis section to explain how ILI episodes were counted and registered (lines 239-246).
Comment 5: In the paper, the experimental results should be interpreted in medical and biological meaning.
Answer 5: We are sorry but unfortunately, we do not understand this comment.
Comment 6: The authors need to explain how determine the threshold in the Sentinel framework and the Grippenet framework.
Answer 6: Thank you for this question. For the Sentinel framework, all we were able to know from the Swiss Federal Office for Public Health is that the moving epidemic method is used on the data from the 10 previous seasons to calculate the threshold. This was added in the introduction, on line 62 of the Track changes version. The explanation about how this framework could be used for Grippenet is on lines 570-578 in the discussion.
Comment 7: There are lots of errors, for examples, (p42 : systems exists >> systems exist, p95 : help seeking >> help-seeking, p107 : for instance >> for instance, p113 : advertisement >> advertisements, p114 : are >> is, p152 : was >> were), and so on. There are too much mistake.
Answer 7: Thank you for pointing this out. We proofread the manuscript and corrected mistakes.
Comment 8: As a monitoring framework for infectious diseases that has recently become an issue, this paper proposes a new tool called Grippenet. Comparing the advantages and disadvantages of various aspects with the existing Sentinel used. However, the comparative analysis of the monitoring framework presented in this paper only compares the results, and users are not sure what principles these two frameworks move on. For example, other existing infectious disease surveillance frameworks are written how monitoring is conducted by some statistical methods or ml-based method. But, this paper simply compares the differences of existing frameworks as results. (A Simulation-Based Study on the Comparison of Statistical and Time Series Forecasting Methods for Early Detection of Infectious Disease Outbreaks)
Answer 8: Thank you for this comment. Indeed, the goal of our article is to compare the Grippenet system with the reference sentinel system. Simulation studies are outside the scope of this work. However, we take your point that the principles these two frameworks move on were not explained clearly enough. We added further explanations about the Grippenet and Sentinella framework in the introduction and further comparison between the two in the discussion (lines 384 - 397)
Comment 9: Also, in the description of the data, how the data was organized is generally detailed, but only fragmentary information reader can get in the actual text. Of course, even if the author has attached the text to the appendix, some of them should be shown to make the paper more credible. This is the same with the problem of feature selection.
Answer 9: Thank you. We added a short description of the questions in each “intake” and “weekly” questionnaires, so that readers do not have to look at the annexes in order to understand what questions were asked to participants.
Comment 10: As described above, it is necessary to describe how the threshold is set up and how the comparison is different from the proposed method. Otherwise, this paper is simply a framework on property extraction, data processing.
Answer 10: see answer 6.
Comment 11: The title also describes influenza-related analysis tools, with data focused on ILI, not influenza. If you want to use this paper as its title suggests, even if the results are lower than the comparison tool, it would be good to add influenza-related results and add descriptions of those results.
Answer 11: Thank you for this comment which points out an important issue, which is why we present you the new title of the manuscript. “Grippenet: a new tool for the monitoring, risk-factor and vaccination coverage analysis of influenza-like illness in Switzerland”.
Comment 12: In addition, comparisons were simply compared in a way that compares the actual situation with previous data, and it seems a little more desirable to compare the performance between the two tools by marking them as a more academic indicator (error rate or accuracy).
Answer 12: Thank you, we added information about the sensitivity and specificity of the ILI definition we used in the methods as well as in the discussion, in order to compare the performance of both systems. We also added information about the percentage of real influenza cases in the Sentinella framework, something that to date cannot be calculated for Grippenet.
Comment 13: Finally, the abstract as summary of the paper as well as most parts in the manuscript should be rewritten.
Answer 13: Replying to the comments here as well as proofreading the abstract and articles, we made many changes that we hope you will find satisfactory.
Reviewer 2 Report
Dear Sir or Madame,
thank you for the opportunity to review the manuscript. Attached you will find my comments:
General comments:
This study presents an internet-based participatory surveillance of influenza-like-illness, complementary to that of an existing sentinel system. The number of Grippenet active users varied according to the incidence of influence like illness (ILI). Factors associated with ILI were: ages, sex, respiratory allergies, vaccination status and self-employment.
The article makes a very valuable contribution to the influenza syndromic surveillance in Europe. It is well written and the described tool offers great opportunities for future development to monitor epidemics like influenza or even Covid-19. The description of the methods may need some more details to avoid misunderstanding and enhance transferability of data.
I would recommend: ‘minor revision’.
- Please clarify within the methods section, how participants were invited to the study. Did they get any kind of incentive?
- Additionally, it would be interesting to know more about the online questionnaire. Did the authors use a validated instrument? Has the questionnaire been used already for another research question / population / health care system?
Thank you very much and best regards!
Author Response
Dear Reviewer,
Thank you very much for your valuable comments on our article originally entitled “Grippenet: a new tool for the monitoring, risk-factor and vaccination coverage analysis of Influenza in Switzerland” (vaccines-805609) submitted as an original article to Vaccines.
We value this opportunity to improve our manuscript and to answer your comments. We used the track change function in Microsoft Word to make the revisions clearly visible.
We thank you for your consideration and look forward for our collaboration.
Reviewer 2’s comments
Comment 1: Please clarify within the methods section, how participants were invited to the study. Did they get any kind of incentive?
Answer 1: Thank you for this comment. We reorganized the methods section by moving the participants’ recruitment subsection at the top for more clarity and expanding it. Also, we added the indication that participation takes place on a voluntary basis with no financial compensation.
Comment 2: Additionally, it would be interesting to know more about the online questionnaire. Did the authors use a validated instrument? Has the questionnaire been used already for another research question / population / health care system?
Answer 2: Thank you for this remark. The questionnaire was developed for the Flusurvey study, which is also part of the Influenzanet consortium. We added more detailed information about the questionnaire questions (lines 155-158 and 165-167) and development (lines 176-180) and the data retrieving process (lines 181-185).
Round 2
Reviewer 1 Report
The answers were not enough for reviewer's comments. Of course, most of comments have been fixed but not all. For example, as a point of view of medical doctor or biologist, the results should be interpreted.
Author Response
Thank you for your review
Comment 1: The answers were not enough for reviewer's comments. Of course, most of comments have been fixed but not all. For example, as a point of view of medical doctor or biologist, the results should be interpreted.
Answer 1: Thank you for this comment. In the authors of our manuscript, we have four physicians, one biologist and one pharmacist. We reworked the paper to add more links to clinical medicine. For example, on lines 56-70 and 158-161 we added more explanations on the relationship between ILI monitoring and real influenza cases, its pitfalls and benefits.
On lines 93-100 we tried to better explain how the project can advance health literacy in the population. We added some country-wide statistics to allow for comparison with the participants’ location in the results, section 3.1. We also further improved clarity, structure and use of English throughout the whole manuscript.